# Development and Validation of the COVID-19-Related Stigma Scale for Healthcare Workers (CSS-HCWs)

**DOI:** 10.3390/ijerph19159641

**Published:** 2022-08-05

**Authors:** Makoto Tsukuda, Yoshiyasu Ito, Keisuke Nojima, Tomonori Kayano, Junko Honda

**Affiliations:** 1College of Nursing Art and Science, University of Hyogo, Kobe 673-0021, Japan; 2Faculty of Nursing, Kyoto Tachibana University, Kyoto 607-8175, Japan; 3Department of Human Nursing, Sonoda Women’s University, Amagasaki 661-8520, Japan

**Keywords:** COVID-19, stigma scale, healthcare workers, reliability, validity, Japan

## Abstract

Stigma among healthcare workers during the coronavirus disease 2019 (COVID-19) pandemic is an issue that requires immediate attention, as it may otherwise lead to the collapse of healthcare systems. In this study, we developed the COVID-19-related stigma scale for healthcare workers (CSS-HCWs) and assessed its reliability and validity. Data were collected online from 500 participants, including physicians and nurses involved in COVID-19 care. The first item of the draft scale was developed based on a literature review and qualitative study. The draft scale consisted of 24 items, which were rated on a six-point Likert scale. Descriptive statistics were calculated and the data distribution was analyzed. To assess the scale’s validity and reliability, structural validity was evaluated through an exploratory factor analysis. Criterion-related validity was examined through a correlation analysis using the E16-COVID19-S, a COVID-19 scale developed for physicians in Egypt. Reliability was evaluated by examining the scale’s stability and internal consistency. The findings revealed that the stigma scale was a valid and reliable instrument. The final scale consisted of 18 items across three domains: personal stigma, concerns of disclosure and others, and family stigma. In conclusion, the scale is a valid and reliable instrument that can measure COVID-19-related stigma among healthcare workers.

## 1. Introduction

The coronavirus disease 2019 (COVID-19) pandemic underscores the enduring threat of infectious diseases to humanity, calling for effective global efforts and a high level of preparedness against future outbreaks [1]. Currently, there are insufficient data to establish a definitive treatment approach with antimicrobials or immunomodulators [2]. Studies suggest that outbreaks of emerging infections, such as COVID-19, have a considerable impact on patients’ mental health, with those affected reporting anxiety symptoms, fear, loss of hope for treatment, and uncertainty about their health status [3]. Several factors affect the mental health of patients with infectious diseases; these include isolation after being diagnosed, stigma and discrimination from others, prolonged hospitalization, and a lack of social support [4]. Stigma not only creates discriminatory behavior and estrangement from family and friends, but also contributes to personal problems, such as negative self-image perceptions. COVID-19 stigma can be understood as a social process that seeks to exclude people who are potential infectious agents and may threaten social life [5].

Furthermore, owing to the COVID-19 pandemic, many countries have reported discrimination against people of Asian descent [6]. As many aspects of COVID-19 are still unknown, there is much confusion, anxiety, and fear among the general public, which can fuel dangerous stereotypes [7]. Among these situations, stigma against healthcare workers has been reported to occur, especially in areas where COVID-19 is considered to be under control [8]. Numerous incidents of violence, harassment, and stigma were reported in connection with the COVID-19 epidemic, 67% of which were directed at patients as well as healthcare workers [9].

Interestingly, stigma about COVID-19 is also prevalent among the same Asian populations, with healthcare workers involved with COVID-19 patients being targeted most frequently [10,11]. Particularly, in Japan, stigmatizing attitudes are stronger than in other countries due to institutionalism, a lack of a national movement to address stigma, and the value society places on conformity [12]. Harassment and stigma related to COVID-19 have been reported in 21.3% of medical facilities in Japan [13]. To develop supportive measures for healthcare workers, it is necessary to determine the causes of stigma and construct a scale that can accurately assess stigma.

Over the years, many instruments to assess stigma, particularly toward severe infectious diseases such as human immunodeficiency virus (HIV) and severe acute respiratory syndrome (SARS), have been developed. The HIV stigma scale developed by Berger [14] has been used in numerous studies to clarify the psychological effects of stigma toward patients with HIV [15,16]. It consists of 40 items across four categories: personalized stigma, disclosure concerns, negative self-image, and concern with public attitude toward individuals with HIV. Social stigma toward HIV and other infectious diseases has detrimental effects on several health-related outcomes. For example, HIV-related stigma is associated with severe anxiety and depression [17]. Pandemic-related work conditions are associated with adverse mental health and intention to leave the workplace [18]. Thus, most studies have assessed the psychological burden of stigma on infected patients. However, stigma toward patients with infectious diseases affects not only the infected persons, but also healthcare workers who come in contact with such patients. Recently, stigma against healthcare workers occurred during the SARS pandemic in 2002 [19,20]. During the 2002 SARS pandemic, healthcare workers who worked with patients in the United States and China experienced stigma and fear, and the importance of psychological support was evident. However, to the best of our knowledge, no previous study has quantitatively measured the stigma experienced by healthcare workers, but rather assessed it qualitatively through interviews, or, alternatively, used the HIV stigma scale [14] mentioned above. A previous study focused on people affected by various infectious diseases [5]. These studies indicate that stigma associated with human immunodeficiency virus (HIV), hepatitis C virus (HCV), tuberculosis, and Zika has an important impact on both the mental and physical health of patients with infectious diseases. However, despite the facts that frontline healthcare workers are at high risk for anxiety, fear, and stress [21] and that they play an indispensable role in healthcare, limited studies have focused on the physical and psychological effects of stigma on frontline healthcare workers.

COVID-19-related stigma has spread worldwide like never before due to the virus’ high infectivity, long incubation period, and the emergence of new mutant strains. Moreover, since it is difficult to easily determine who is infected, not only healthcare workers caring for COVID-19 patients, but also those not caring for COVID-19 patients and their families are subject to stigmatization [22]. A qualitative study of COVID-19-related stigma for nurses conducted in Japan reported that one of the components of stigma was “discrimination toward family members”, which is a characteristic of COVID-19-related stigma [23]. Several measures of COVID-19-related stigma for healthcare workers have been used to measure it [24,25], but these do not reflect other characteristics of COVID-19-related stigma such as the stigmatization of family members of healthcare workers. To continue providing healthcare services to patients with COVID-19, validated measures to assess healthcare workers’ psychological burden play a major role in assisting these professionals with the mental health and psychological aspects of the disease [26]. Therefore, in this study, we aimed to develop the COVID-19-related stigma scale for healthcare workers (CSS-HCWs) and to examine its reliability and validity.

## 2. Materials and Methods

### 2.1. Study Design

The scale development was based on the guidelines described in Scale Development: Theory and Applications [27] and the Consensus-based Standards for the Selection of Health Status Measurement Instruments checklist [28].

### 2.2. Setting and Participants

Participants were physicians and nurses working in hospitals that accepted COVID-19 patients. We targeted physicians and nurses who had worked with COVID-19 patients, as they would have more experience with COVID₋19-related stigma. At the time of the study, Japan was at emergency alert level 2, which refers to a situation where the number of infected people is decreasing, and social activities are conducted while infection prevention actions are implemented. In addition, Japan had prohibited the entry of new foreigners into the country.

To meet the criteria for good methodology, a sample size of at least seven times the number of survey items and an absolute number of at least 100 were required [28]. The first version of the questionnaire had 40 items, so the minimum sample size was 280. This survey was commissioned by Rakuten Insight, Inc. (https://insight.rakuten.co.jp/en/ accessed on 9 June 2022) and was conducted via an internet-based questionnaire. Survey requests were sent to monitors (doctors and nurses) registered with Rakuten Insight and who met the selection criteria. The inclusion criteria for participants were: physicians and nurses working at the hospital on a full-time basis. Licensed practical nurses/nursing assistants and midwives were excluded. Of the monitors to whom the survey request was sent, those who were willing to participate in the survey were selected, and the sampling continued until 250 physicians and 250 nurses were included. To verify data collection, including reproducibility, two surveys were conducted to examine reliability through a test–retest. Two weeks after the first survey, the second survey was administered to all participants of the first survey.

### 2.3. Instruments

Participants answered a questionnaire consisting of three sections: (1) participants’ demographic characteristics, (2) the CSS-HCWs, and (3) the Japanese version of the 16-item COVID-19 stigma scale (E16-COVID19-S).

#### 2.3.1. Participant Demographics

Data on participants’ age, gender, clinical career (years of work experience), education level, family type, and whether they were directly involved with COVID-19 patients were collected.

#### 2.3.2. CSS-HCWs

To investigate participants’ stigma-related thoughts and behaviors, we examined the mental stress caused by stigma and investigated existing measures of stigma toward patients with infectious diseases, such as HIV and SARS, and healthcare professionals through a literature review [29]. In addition, because COVID-19 is an emerging infectious disease and effective treatment methods have not yet been established, we sought to understand participants’ stigma-related thoughts and behaviors caused by infectious diseases that were already under control. Based on the keywords “avoid”, “hide”, “pry”, “guilt”, “fear”, and “repression” extracted from a literature review of previous studies, we conducted in-depth interviews and developed questionnaire items about the stigma perceived by healthcare workers working in hospitals that provided care for patients with COVID-19. From the interviews, keywords such as “family stigma”, which is considered a characteristic of stigma against healthcare workers, and “perceive” and “do not express”, which are considered cultural characteristics of Japan, were added to the candidate items, which were developed to take these characteristics into account [23]. Items were reviewed by several experts in infectious diseases to assess the content validity, and some items were deleted or added. The resulting 24 items were administered to the respondents as a draft questionnaire. This was a six-point Likert-type scale; that is, there were six options for each item, “do not agree at all”, “do not agree”, “do not agree very much”, “somewhat agree”, “agree”, and “very much agree”, corresponding to 1–6 points, respectively.

#### 2.3.3. Japanese Version of the E16-COVID19-S

To assess the criterion-related validity, the E16-COVID19-S was used. It is a COVID-19 scale developed for physicians in Egypt, based on the SARS and HIV stigma scales, and consists of 16 items across three factors: personalized stigma, concerns of disclosure and public attitudes, and negative experiences. The Cronbach’s α coefficient was 0.909. E16-COVID19-S was confirmed to have a good model fit (comparative fit index = 0.964; root mean squared error of approximation = 0.056). The scale’s reliability and validity were confirmed [25]. After obtaining permission from the scale developer, the E16-COVID19-S was translated by two native Japanese speakers who had a good command of English. A common version was developed in English by synthesizing the two translated versions. The translated version was retranslated by two independent native English speakers who were fluent in Japanese and who were not involved in the initial translation process (from Japanese to English). The original and retranslated versions of the E16-COVID19-S were compared, and the Japanese version was developed. The scale was a six-point Likert-type scale; that is, there were six options for each item, “do not agree at all”, “do not agree”, “do not agree very much”, “somewhat agree”, “agree”, and “very much agree”, corresponding to 1–6 points, respectively. The internal consistency of the E-16 COVID-19-S Japanese version had a Cronbach’s α coefficient of 0.981.

It is recommended that a newly developed questionnaire be pilot tested and peer-reviewed [27]. Therefore, prior to the main survey, we administered a pretest to 10 healthcare workers as a face validity test. Participants were asked the following questions: “Are the instructions clear and easy to follow?” and “Is it easy for you to understand the questions and choose your answers?”. Feedback from the participants was mainly related to the phrasing of some questions in Japanese, which we modified to avoid any misunderstandings. All coauthors and several experts evaluated the questionnaire to ensure that the items were appropriate and that the questions were not difficult for participants to answer, and so, the questionnaire was finalized.

### 2.4. Data Analysis

In the statistical analysis, an item analysis was performed first. An exploratory factor analysis (EFA), confirmatory factor analysis (CFA), and criterion-related validity were performed to assess the scale’s validity. Internal consistency and reproducibility were examined to assess reliability. IBM SPSS (version 28.0, SPSS Inc., Chicago, IL, USA) and AMOS (version 28.0, Europress, Adlington, UK) were used for the analyses.

#### 2.4.1. Item Analysis

Standard deviations and means of all items estimated to detect ceiling and floor effects were found to be in the 1–6 range. The item–total (I–T) and good–poor (G–P) correlations were examined.

#### 2.4.2. Validity Analysis

An EFA was performed to determine a factor structure that supported COVID-19-related stigma perceived by healthcare professionals. The Kaiser–Meyer–Olkin (KMO) test to confirm sampling adequacy and Bartlett’s specificity test were performed to assess the fit of the data to the factor analysis [30]. The number of factors was visually inspected using scree plots to identify the ideal number of potential factors and examine whether it met the KMO criteria [31] of eigenvalues ≥ 1 and the proportion of variance explained by each factor [31]. A maximum likelihood (ML) method with promax rotation was then performed [32]. Items with factor loadings < 0.5 were excluded. As a rule, the total variance explained should be >50% [33]. Goodness of fit of the factor structure of the CSS-HCWs was tested using CFA. The chi-squared test, goodness of fit index (GFI), comparative fit index (CFI), and root mean square error of approximation (RMSEA) were used to assess data fit. The goodness of fit of the model was considered very good when GFI > 0.95, CFI > 0.90, and RMSEA < 0.05. The ratio of chi-squares with respect to the degrees of freedom (χ2/df < 2.00) was also evaluated. For criterion validity, Spearman’s correlation coefficient was used to evaluate the correlation between the proposed scale scores and the E16-COVID19-S scores.

#### 2.4.3. Reliability Analysis

The entire scale and each subscale were analyzed using Cronbach’s alpha coefficient to confirm the reliability of the scale. In addition, to verify the test–retest reliability, we determined the intraclass correlation coefficient of the response scores to the stigma scale obtained at the two time points [34].

### 2.5. Ethical Considerations

Potential participants were provided with a written request for participation, explaining the purpose and methods of the study and information about the confidentiality of their data, the voluntary nature of participation, the stringent protocols for storage and disposal of personal information, and the publication of the study results. Only those who agreed to the study were administered the questionnaire. Data were collected anonymously using serial IDs, and data confidentiality was ensured. This study was approved by the ethics committee of the university to which the authors were affiliated (approval number 2021F14).

## 3. Results

### 3.1. Participant Characteristics

Participants included 250 physicians and 250 nurses. Regarding gender, 267 were male and 233 female. The mean scores of the CSS-HCWs (max: 108; min: 18) were 49.65 and 51.51, respectively, with no significant differences. The mean clinical experience was 18.4 ± 10.8 years (range: 0–43 years). There were 372 participants (74.4%) who lived with family members and 128 (25.6%) who did not live with family members. The CCS-HCWs scores were 50.15 and 51.84, respectively. There were no significant differences (Table 1).

### 3.2. Evaluation of the Reliability and Validity of the CSS-HCWs

#### 3.2.1. Answer Distribution and Item Analysis

Data from 500 healthcare workers were used for an item analysis. Most items had the highest frequency of “three-point” responses. Items 9 and 10 had the highest frequency of “one-point” responses. However, none of the items had a response frequency above 90%. The maximum mean and standard deviation for each item was 4.667 and the minimum was 1.159, with no ceiling or floor effects. I–T correlations were significantly correlated with the overall scores for all items. Thus, no items were removed from the item analysis.

#### 3.2.2. Validity Testing

Construct validity: An item analysis was performed on the 24 items of the proposed scale; no items were deleted. Therefore, an EFA was performed on all 24 items using the ML method and promax rotation. The KMO index of sampling adequacy was 0.955. Based on eigenvalues ≥ 1.0 and scree plots, the number of factors was three. Items with loading values ≥ 0.5 were retained. Six items (2, 4, 11, 17, 18, and 20) were removed because their factor loadings were <0.5. Finally, an 18-item scale with a three-factor structure was employed. Each factor was interpreted and named by comparing the theoretical characteristics derived from the three-factor structure employed in the EFA and the constructs identified in a previous study [23].

The following types of stigma were identified as relevant to healthcare workers:(1)“Personal stigma” reflected what healthcare workers perceived as a result of their close contact with COVID-19 patients.(2)“Concerns of disclosure and attitude” reflected concerns about their status as a healthcare worker and their close contact with COVID-19 patients.(3)“Family stigma” reflected the concern that one’s family would be treated unfairly because of their status as a healthcare worker and their close contact with COVID-19 patients.

Factors 1–3 explained 64.52%, 5.334%, and 3.538%, respectively, of the total variance, and the cumulative contribution rate was 73.400% (Table 2). The CFA for this 18-item model yielded the following indices: χ2 (126) = 492.714; χ2/df = 3.910; GFI = 0.902; CFI = 0.962; and RMSEA = 0.076.

Criterion-related validity: Table 3 presents the results of the criterion-related validity assessment, which involved examining the extent to which the total score on the stigma scale and the scores on each factor correlated with the total score on the E16-COVID19-S. Spearman’s rank correlation coefficient for the total score was 0.875, with values for factors 1–3 ranging from 0.641 to 0.851 (*p* < 0.01).

#### 3.2.3. Reliability Testing

Internal consistency: An analysis of the internal consistency of the final three-factor, 18-item scale showed high reliability for all factors. The Cronbach’s α coefficient for the overall scale was 0.970. The coefficients for the subscales “personal stigma”, “concerns of disclosure and attitude”, and “family stigma” were 0.940, 0.943, and 0.955, respectively.

Test–retest reliability: Of the 500 respondents who responded to the first survey, 460 (92.0%) participated in the second survey. The intraclass coefficient for the test–retest reliability was 0.836 (*p* < 0.01) for the overall scale. The coefficients for the subscales “personal stigma”, “concerns of disclosure and attitude”, and “family stigma” were 0.814 (*p* < 0.01), 0.814 (*p* < 0.01), and 0.795 (*p* < 0.01), respectively.

## 4. Discussion

In this study, the CSS-HCWs was developed by employing a comprehensive approach. The initial items were developed through a qualitative study on healthcare workers involved with COVID-19 patients, and it included the participants’ perspectives on cultural norms and values. This approach was considered the basis for tool development in psychometry [35]. Meanwhile, the conceptual factor structure (construct validity) was assessed by the EFA. The developed scale had no floor or ceiling effects. The results indicated that social stigma among Japanese healthcare workers was best explained by the three-factor, 18-item model. To our knowledge, this is the first COVID-19-related stigma scale that assesses the psychological burden of Japanese healthcare workers. In this study, physicians and nurses, who were expected to feel considerable COVID-19-related stigma, were included. We found no difference in the perceived stigma scores on the CSS-HCWs between the physicians and nurses, suggesting that the scale could be used to assess self-stigma and perceived stigma among various healthcare professionals.

Cronbach’s alphas exceeded 0.80 for the overall scale and the three factors, indicating sufficient reliability [36]. Thus, the internal consistency of the CSS-HCWs was verified. Regarding the test–retest reliability, the overall score on the stigma scale and the intraclass correlation coefficients for the factor scores of respondents’ first and second responses were strongly correlated [37], indicating the stability of the scale. Regarding construct validity, the results of the EFA revealed a three-factor structure, comprising a new family-related category identified in this study and two stigma-related categories defined in a previous study [23]. Thus, this scale is considered to be a good tool for measuring stigma, with verified reliability and validity.

In the qualitative survey [29], two themes were derived: directly experienced prejudice and discrimination (“being avoided”, “being treated as dirty”, ”discrimination toward family members”, ”prying questions from others”) and self-imposed coping behavior (“keeping to oneself”, ”feeling guilty”, ”non-disclosure”).

Further, in the CSS-HCWs, family-related content was an independent factor. In stigma scales created for healthcare professionals in India and Egypt [25,38], family-related content was included within other factors, or family-related items were not included at all. Stigma commonly refers to self-stigma and social stigma. In addition to these forms of stigma, stigma toward family was added as a new factor in the CSS-HCWs. This may be explained by the prominent emphasis on family values in Japan, the cultural characteristics of the Asian region, and a strong sense of responsibility that is unique to healthcare workers [12].

Providing emotional support to healthcare workers can increase productivity and decrease turnover rates among healthcare workers. The World Health Organization supported this in its 2019 report, noting that stigma, as well as harassment at the workplace, can negatively affect physical and mental health, which may be reflected in higher staff turnover and lower productivity levels [39]. In addition, studies focusing on organization-level interventions that promote mental health at the workplace found a correlation between mental health and the well-being of healthcare workers. A continuous assessment and intervention approach to improve the mental health of medical staff is needed at the organization level. Interventions to reduce COVID-19 stigma are recommended in all countries. Stigma is thought to manifest across cultures, and effective interventions to reduce it are thought to have similar basic characteristics [40]. However, quantitatively measuring the content and intensity of any stigma with respect to specific, effective interventions and their evaluation is required. By utilizing the CSS-HCWs developed in this study, we believe that it can be possible to present specific numerical values of what stigma HCWs feel and to what extent.

Future research should consider using the scale developed in this study for specific interventions and evaluations to reduce stigma and improve the mental health of healthcare professionals.

### Study Limitations

There was a possibility of selection bias due to the facts that this study was conducted online and that the participants were limited to those registered with Rakuten Insight. The goodness of fit of this model might not be adequate, but we believe that this value was within an acceptable range. We also believe that the adoption of stigma items related to family were extracted from a qualitative survey. Since this study’s sample comprised physicians and nurses in Japan, its applicability to other countries and other healthcare professionals must be examined.

## 5. Conclusions

The CSS-HCWs is a reliable and validated instrument for quantitatively assessing stigma in healthcare workers caring for COVID-19 patients. It is also the first instrument to be developed with fidelity from qualitative data from healthcare providers who actually provided care to COVID-19 patients.

Healthcare workers have a special sense of mission and responsibility that is different from that of the average person. Therefore, the felt stigma may not be outwardly expressed, but internalized, leading to a psychological burden. By utilizing the CSS-HCWs developed at the time, the stigma felt by healthcare professionals can be broadly classified into “personal stigma”, “concerns of disclosure and attitude”, and “family stigma”, and its magnitude can be measured and classified.

If common support measures for stigma and specific support measures for healthcare workers can be systematically implemented, it would be possible to effectively intervene and evaluate stigma against emerging infectious diseases that may occur in the future.

## Figures and Tables

**Table 1 ijerph-19-09641-t001:** Participant characteristics (*n* = 500).

Characteristics	Categories	*n* (%)	Mean ± SD	Total Score
Sex	Male	267		49.58
	Female	233		51.73
Occupation	Physician	250 (50.0)		49.65
	Nurse	250 (50.0)		51.51
Clinical experience (years)	0–9	126		48.6
	10–19	149		51.09
	20–29	127		51.8
	30–39	86		51.13
	≥40	12		48.42
			18.4 ± 10.8	
Educational background	Vocational school	151 (30.2)		53.76
	University	223 (44.6)		50.12
	Graduate school	126 (25.2)		48.83
Living with family members	Yes	372 (74.4)		50.15
	No	128 (25.6)		51.84

**Table 2 ijerph-19-09641-t002:** Results of the exploratory factor analysis and Cronbach’s α coefficients (*n* = 500).

Item			Factor Loading		
			1	2	3
Factor 1: Personal Stigma (α = 0.940)					
7	My family would treat me as dirty if I have close contact with COVID-19 patients.		0.938	−0.058	−0.043
5	Other healthcare workers would treat me as dirty if I have close contact with COVID-19 patients.		0.803	0.046	0.055
3	My family would avoid me if I have close contact with COVID-19 patients.		0.777	−0.166	0.182
9	I feel guilty if I have had close contact with COVID-19 patients.		0.761	0.152	−0.114
1	Other healthcare workers would avoid me if I have close contact with COVID-19 patients.		0.649	−0.039	0.284
12	If healthcare workers get infected with COVID-19, it is no surprise that they are criticized by the others.		0.623	0.123	−0.086
8	I find myself feeling dirty if I have had close contact with COVID-19 patients.		0.547	0.201	0.015
10	People treat me as a bad person if I have close contact with COVID-19 patients.		0.537	0.196	0.169
6	People would treat me as dirty if I have close contact with COVID-19 patients.		0.536	0.098	0.303
Factor 2: Concerns of Disclosure and Attitude (α = 0.943)					
23	I should hide the fact that I am a healthcare worker.		0.039	0.921	−0.077
22	It is best to hide the fact that I have had close contact with COVID-19 patients.		−0.011	0.819	0.096
24	I ask my family to hide the fact that I am a healthcare worker.		0.127	0.808	−0.032
21	It is dangerous to tell people that you have had contact with COVID-19 patients.		−0.126	0.754	0.284
19	It is better not to meet with people around you so that they do not ask whether you have had close contact		0.197	0.572	0.162
Factor 3: Family Stigma (α = 0.955)					
13	The families of healthcare workers who have close contact with COVID-19 patients will be treated unfairly at work and school.		0.008	0.053	0.900
15	The families of healthcare workers who have close contact with COVID-19 patients will be avoided by people.		0.006	0.047	0.895
14	The families of healthcare workers who have close contact with COVID-19 patients will be treated as an infected person.		0.039	0.090	0.815
16	The families of healthcare workers who have close contact with COVID-19 patients will interfere with work and schoolwork.		0.138	0.097	0.682
Factor loading (%)			64.52	5.334	3.538
Cumulative loading (%)				69.862	73.400
Cronbach’s α (full scale) = 0.970					
Interfactor correlations		Factor 1	1.000	0.722	0.776
		Factor 2		1.000	0.769
		Factor 3			1.000
Excluded Items					
2	I will be avoided by people because I had close contact with COVID-19 patients.				
4	I will avoid people because I had close contact with COVID-19 patients.				
11	If healthcare workers get infected with COVID-19, some people treat me like a bad person.				
17	Some people pry I had close contact with COVID-19 patients.				
18	I don’t hate being asked if I had close contact with COVID-19 patients.				
20	Some people pry if I had close contact with COVID-19 patients through my family.				

**Table 3 ijerph-19-09641-t003:** Relationship between CSS-HCWs and E16-COVID19-S (*n* = 500).

	CSS-HCWs			
		Factor 1	Factor 2	Factor 3
E16-COVID19-S	Total score	Personal stigma	Concerns of disclosure and attitude	Family stigma
Total score	0.875 **	0.837 **	0.839 **	0.729 **
Personalized stigma	0.856 **	0.836 **	0.806 **	0.710 **
Concerns of disclosure and public attitudes	0.860 **	0.795 **	0.851 **	0.732 **
Negative experiences	0.797 **	0.784 **	0.753 **	0.641 **

Note. Spearman’s rank correlation coefficient for the total score between CSS-HCWs and E16-COVID19-S. ** *p* < 0.01.

## Data Availability

Not applicable.

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
