# Peer review of "Development and Validation of the COVID-19-Related Stigma Scale for Healthcare Workers (CSS-HCWs)"

_ijerph, 2022, doi:10.3390/ijerph19159641_

Round 1

Reviewer 1 Report

1. Introduction: The reason for the strong stigma in Japan was explained by the social atmosphere, and the stigma is thought to affect the level of quarantine. I think it would be good to explain the level of quarantine in Japan at the time of the study and the level of quarantine. This should also be mentioned in the limitations.

2. Materials and Methods: What is the response rate and rewards for participating in surveys?

3. Results: The gender of the subject is not mentioned, has it not been investigated? If so, the limitations need to be mentioned.

4. Discussion: The strength of the developed tool is to measure the stigma of the family. In order to resolve this stigma, it would be good to suggest specific solutions (social systems, laws, awareness improvement, etc.) other than emotional support.

Reviewer 2 Report

Comments and suggestions for authors

I read with interest the manuscript entitled ‘Development and Validation of the COVIDâ‚‹19-Related Stigma Scale for Health Care Workers (CSS-HCWs)’ and I appreciate the work of the authors.

The authors highlighted the importance of phenomenon of stigmatization of health workers in the context of Covid-19 pandemic. Given the effects and consequences of this phenomenon, the authors have taken a step forward and developed a questionnaire that can assess the stigma that health workers and their families may face.

The manuscript is clear and presented in a well-organized manner. 

Introduction is well written, but the authors should cite more recent references.  

Materials and methods

The authors should specify the selection criteria (lines 100-101: ‘Survey requests were sent to monitors …who met the selection criteria’).

The authors have conducted a literature review (lines 119, 123), but nothing is specified in this regard.  Although there may be several articles/ papers, the authors should consider whether some of them should still be mentioned in the text and in the references’ list.

The statement regarding ‘the keywords which are considered cultural characteristics of Japan’ it would be useful to be supported by specific references, especially since the indicated reference [29] is, at this moment, in press.

The research design followed the methodology for a scale development.

Results and Discussions are clearly presented.  

At lines 288 -the authors refer to the scales developed in Egypt (about which it was written above) and in India. The latter appears only now in the text and a reference should be added.  

References

Many cited references are very current.  However, out of a total of 43 references, 18 are prior to 2017.  Some of them [16, 18-23] are mainly related to HIV patients; perhaps the authors will review these references, especially since the topic of the article focuses on health workers. The authors should improve this aspect.

Other observations

 Lines 109: It would be useful to mention what ‘E16-Covid19-S’ represents (it is the first time used in the text).

 Lines 132-134 and 148-151 are identical. The authors should check if this repetition is necessary.

Reviewer 3 Report

Dear Authors,thanks for sharing your experience. The impact of the current pandemic, in fact, is transversal to human nature itself, affecting health and well-being at all levels. Even if it's interesting, your contribution must be improved. Here some insights
  • Introduction:
  •  Regarding Introduction, I would like to say that the first 30 lines are in my opinion very anachronistic. COVID 19 is noo longer the Chinese virus, instead China has adopted a zero-COVID policy and right now the current COVID variants (which are really different from the Wuhan virus) developed in other countries than China. The Chinese virus and the dimiscrimination against Eastern people belong to the past; so I highly recommend to fix this initial part and update it. . Also I recommend to better constestualize stigma against workers and esepcially HCWs.
    • contextualize the pandemic scenario underway, with clear references to the international (WHO), supranational and national entities involved in the management of the emergency, reporting official data (i.e. John Hopkins University observatory)
    • contextualize the stigma and discrimination phenomenon, not simply recalling HIV
    • why not SARS-CoV-2 among keywords?
  • Material and methods:
    • questionnaire administered is tailor-made, right? Does it contain parts of validated questionnaires? Please provide detailed informations
    • please provide ethical committee approval and statement
    • ... "was commissioned by Rakuten Insight, Inc" but, in funding statement you declare "Funding: This research was funded by Foundation of Kinoshita Memorial Enterprise."; please clarify and adjust this point
    • why did you just stopped the recruitment at 500 total participants? How many were really invited? You can't just state the final number of participants. That's possibly another bias that sums to other you have to discuss about
  • Discussion/Conclusion:
    • SES is double-stranded to work conditions and job security; you've just grazed it, why?
    • deepen discussion about gender
Discussion can be improved
Conclusion must be improved. What are the possible repercussions? What suggestions to give to the health policy maker? Define a clear "take home message" from your perspective and address a conclusion section. You need conclusions."The CSS-HCWs is a reliable and validated instrument for quantitatively assessing 312 stigma in health care workers caring for COVID-19 patients." Are you sure you can state this phrase? Please state in the conclusion if you will really attempt to validate your questionnaire.Please update these gaps referring to the following references:
  • https://www.who.int/westernpacific/emergencies/covid-19/information/social-stigma-discrimination
  • Baldassarre, A.; Giorgi, G.; Alessio, F.; Lulli, L.G.; Arcangeli, G.; Mucci, N. Stigma and Discrimination (SAD) at the Time of the SARS-CoV-2 Pandemic. Int. J. Environ. Res. Public Health 2020, 17, 6341
  • M. Tasdik HASAN, Sahadat HOSSAIN, Tanjir R. SARAN, Helal U. AHMED. Addressing the COVID-19 related stigma and discrimination: a fight against “infodemic” in Bangladesh. Minerva Psichiatrica 2020 December;61(4):184-7 DOI: 10.23736/S0391-1772.20.02088-9
  • Gronholm, P., Nosé, M., Van Brakel, W., Eaton, J., Ebenso, B., Fiekert, K., . . . Thornicroft, G. (2021). Reducing stigma and discrimination associated with COVID-19: Early stage pandemic rapid review and practical recommendations. Epidemiology and Psychiatric Sciences, 30, E15. doi:10.1017/S2045796021000056
  • Sotgiu G, Dobler CC. Social stigma in the time of coronavirus disease 2019. Eur Respir J 2020; 56: 2002461 [https://doi.org/10.1183/13993003.02461-2020].
  • Divya Bhanot, Tushar Singh, Sunil K. Verma and Shivantika Sharad. Stigma and Discrimination During COVID-19 Pandemic. Front. Public Health, 12 January 2021 | https://doi.org/10.3389/fpubh.2020.577018
Last but not least:
  • what about vaccine hesitancy in this scenario? (please also refer to https://www.who.int/news-room/spotlight/ten-threats-to-global-health-in-2019 )
Thank you  

Reviewer 4 Report

The authors propose an original scale to evaluate the stigma derived from COVID-19 in health workers and evaluate its reliability and validity. It is a novel work, with contributions to address current social problems.

INTRODUCTION

1. It is necessary to describe studies. Recently, stigma against healthcare workers occurred during the SARS pandemic in 2002 [25, 26]. line 66

2. Missing to mention studies. Previous studies have focused on people affected by various infectious diseases. line 67

MATERIALS AND METHOD

3. Detail how you analyzed the ceiling and floor effects 169

RESULTS

I found it adequate and well presented.

DISCUSSION

I found it appropriate and well presented.

Round 2

Reviewer 3 Report

Dear Authors,

I encourage you to follow my suggestions of the previous report. 

If the scope of the article is the validation of a stigma scale, the Discussion and the Conclusion need to be enriched to justify the need of such a scale. 

Also, in justifying the sample, please add your explanation in the text. 

Again, I think that the construction of the scale needs more clarification.

Best regards
